# Efficient Low-Bit Quantization with Adaptive Scales for Multi-Task Co-Training

**Boyu Liu**[1†]**, Haoyu Huang**[2†]**, Linlin Yang**[3*]**, Yanjing Li**[4*]**, Guodong Guo**[5]
**Xianbin Cao**[4]**, Baochang Zhang**[1,6,7,8]
[1]Institute of Artificial Intelligence, Beihang University
[2]National Superior College for Engineers, Beihang University
[3]State Key Laboratory of Media Convergence and Communication,
 Communication University of China
[4]School of Electronic Information Engineering, Beihang University
[5]Ningbo Institute of Digital Twin, Eastern Institute of Technology, Ningbo
[6]Hangzhou Innovation Institute, Beihang University
[7]Zhongguancun Laboratory [8]Nanchang Institute of Technology

## Abstract

Co-training can achieve parameter-efficient multi-task models but remains unexplored for quantization-aware training. Our investigation shows that directly introducing co-training into existing quantization-aware training (QAT) methods results in significant performance degradation. Our experimental study identifies that the primary issue with existing QAT methods stems from the inadequate activation quantization scales for the co-training framework. To address this issue, we propose Task-Specific Scales Quantization for Multi-Task Co-Training (TSQ-MTC) to tackle mismatched quantization scales. Specifically, a task-specific learnable multi-scale activation quantizer (TLMAQ) is incorporated to enrich the representational ability of shared features for different tasks. Additionally, we find that in the deeper layers of the Transformer model, the quantized network suffers from information distortion within the attention quantizer. A structure-based layer-by-layer distillation (SLLD) is then introduced to ensure that the quantized features effectively preserve the information from their full-precision counterparts. Our extensive experiments in two co-training scenarios demonstrate the effectiveness and versatility of TSQ-MTC. In particular, we successfully achieve a 4-bit quantized low-level visual foundation model based on IPT, which attains a PSNR comparable to the full-precision model while offering a $7.99\times$ compression ratio in the $\times 4$ super-resolution task on the Set5 benchmark.

## 1 Introduction

The paradigm of multi-task training (Caruana, 1997) facilitates knowledge sharing between different tasks and enables the learning of universal representations through multi-objective training across diverse datasets. Multi-task learning has achieved significant advancements in various fields such as natural language processing (Collobert & Weston, 2008; Raffel et al., 2020), computer vision (Eigen & Fergus, 2015; Likhosherstov et al., 2022) and multi-modal learning (Hu & Singh, 2021; Chen et al., 2023a).

Unlike early multi-task learning methods (He et al., 2017; Zhang et al., 2014), which develop models to complete multiple tasks simultaneously, co-training (Maninis et al., 2019) trains a versatile model that performs a single task at a time for a given input. This learning strategy enables the co-trained model to handle multiple datasets with varying characteristics (Zhang et al., 2021). Therefore, co-training has been widely used to establish a universal representation learning framework for multi-modal data (Girdhar et al., 2022) or single-modal task-related data (Chen et al., 2021). No-

---

[†]Equal contribution.
[*]Corresponding Authors: lyang@cuc.edu.cn, yanjingli@buaa.edu.cn.

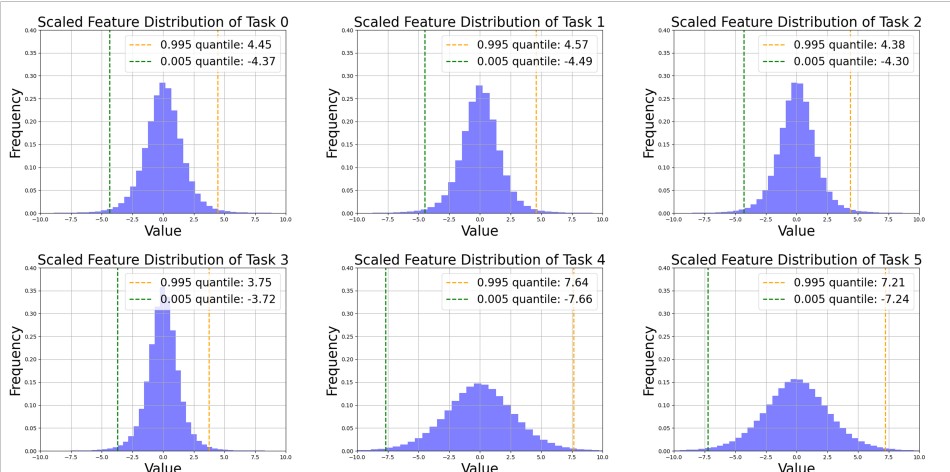

Figure 1: Feature distributions of *decoder.layer9.self_attn.value* in the 4-bit IPT model co-trained with LSQ+ (see **baseline-multi** in Sec. 3.2). The first row shows three tasks (tasks 0, 1, and 2) for $\times 2$, $\times 3$, and $\times 4$ super-resolution tasks, respectively. In the second row, task 3 corresponds to the deraining task, while task 4 and task 5 are denoising tasks with $\sigma = 30$ and $\sigma = 50$. The green dashed line indicates the position of the 0.995 quantile, and the orange one shows the position of the 0.005 quantile. We can observe significant distribution differences between tasks, which require diverse activation quantization scales for co-training.

tably, the co-training paradigm efficiently integrates multiple tasks into a single framework through shared parameters, offering considerable advantages for deployment on edge computing devices (Likhosherstov et al., 2022).

Despite the high performance with shared parameters, the inherent computational and memory overhead of co-trained models still restricts deployments in scenarios with limited memory and computational resources (Likhosherstov et al., 2022). By integrating low-bit quantization (Esser et al., 2019; Hubara et al., 2021) into the co-training framework, we can effectively enhance the deployment potential of co-trained models in practical scenarios. Quantization mainly involves mapping the original floating-point values of weights and activations to lower-bit representations, such as 8-bit or even binary formats, while striving to maintain model accuracy (Zhou et al., 2016). For models designed for image classification and object detection, quantization methods have been extensively studied and shown notable success (Yang & Jin, 2021; Wang et al., 2022; Xu et al., 2023). However, in the context of co-training, directly introducing quantization-aware training (QAT) (Bhalgat et al., 2020) suffers from inferior quantization performance.

In this work, we focus on integrating quantization-aware training with the co-training paradigm, which is underexplored in prior research. Based on the 4-bit QAT baseline established on Image Processing Transformer (IPT) (Chen et al., 2021), we observe that the quantized baseline model after co-training exhibits significant performance degradation compared to its full-precision counterpart, with varying performance gaps across different tasks. Through an analysis of the activation quantizer, we identify that the incompatibility of existing QAT methods for co-training primarily stems from the activation quantization scales. Specifically, as shown in Fig. 1, for the same activation quantizer, the input feature distributions vary significantly between tasks, particularly between super-resolution and denoising tasks. This discrepancy results in a conflict, which degrades the representational ability of the shared quantized features across tasks. Moreover, we find that in the deeper layers of the Transformer model (Vaswani, 2017), the quantized attention module often experiences information distortion. Therefore, simple logit-level supervision is ineffective and coarse-grained for differences in attention information distortion across different layers.

In this paper, we build the baselines by naively integrating the multi-task co-training with QAT, validate their effectiveness, and show the analysis through comprehensive experiments. Based on baselines and analysis, we propose an effective low-bit quantization method, Task-Specific Scales Quantization for Multi-Task Co-Training (TSQ-MTC). To address the issue of mismatched quantiza-

tion scales during co-training, we introduce a task-specific learnable multi-scale activation quantizer (TLMAQ), enabling different task inputs to have richer representational ability and improve the low-bit quantization performance. Meanwhile, to ensure that the quantized features effectively preserve the information from their full-precision counterparts, we propose a structure-based layer-by-layer distillation (SLLD). It leverages the structural similarity of features, thereby enhancing the model's adaptability in each task. We conduct comprehensive experiments in two co-training data scenarios: a single-modal task-related data scenario based on IPT and a multi-modal data scenario based on ResNet, demonstrating the effectiveness of TSQ-MTC.

To summarize, our contributions are as follows:

1. Our Task-Specific Scales Quantization for Multi-Task Co-Training (TSQ-MTC) effectively incorporates quantization-aware training into co-training and significantly reduces the performance gap between multi-task co-trained models and their 4-bit quantized counterparts.

2. We introduce a task-specific learnable multi-scale activation quantizer (TLMAQ) for multi-task co-training. TLMAQ addresses the issue of mismatched quantization scale and improves the representational capabilities of quantized features across tasks.

3. We design a structure-based layer-by-layer distillation (SLLD). It deals with information distortion problems in deeper layers of Transformer, thereby enhancing framework's effectiveness and versatility.

4. Extensive experiments on two co-training data scenarios (a single-modal task-related data scenario and a multi-modal data scenario) show that our 4-bit quantization outperforms the baselines by a large margin and achieves comparable performances with the full-precision counterparts.

## 2 RELATED WORK

### 2.1 MULTI-TASK LEARNING AND CO-TRAINING

Early multi-task learning (MTL) methods (Zhang et al., 2014; Eigen & Fergus, 2015; Ren et al., 2016) improve the performance by using a single input to produce different task outputs. They aggregate the loss of tasks while adjusting their importance. However, these models encountered the challenge of task interference (Maninis et al., 2019), where a potent backbone could enhance overall multi-task performance, yet the performance of each task might still fall short compared to that achieved through single-task training with the same backbone (He et al., 2017; Kokkinos, 2017). Co-Training (Maninis et al., 2019), also referred to as "single-tasking of multiple tasks", develops a model trained on multiple tasks but executes only one task at a time, effectively addressing task interference in shared networks. As research on unified frameworks (Gao et al., 2023; Wang et al., 2023; Han et al., 2024) continues to expand, co-training is typically adopted in two types of data scenarios: the multi-modal data scenario and the single-modal task-related data scenario. In the former scenario, co-training integrates varying modalities of data and tasks to unify representations (Hu & Singh, 2021; Zhang et al., 2021; Likhosherstov et al., 2022; Chen et al., 2023b; Srivastava & Sharma, 2024). In contrast, in the latter scenario, models are trained to focus on the intrinsic correlation of feature learning among different tasks under a single modality (Chen et al., 2021; Ma et al., 2024; Li et al., 2024). Due to the training efficiency and multi-task capabilities of co-training, it is frequently utilized as one of the training techniques for foundation models (Chen et al., 2023a). However, incorporating quantization-aware training into this learning approach to further enhance the model's lightweight deployment capabilities remains an underexplored area for research.

### 2.2 QUANTIZATION SCALE

Quantization scale (the width of a quantization bin) is a crucial parameter within quantizers, enabling effective quantization mappings for both activation inputs and weights (Esser et al., 2019). Early methods (Li et al., 2016; McKinstry et al., 2018) typically calculate quantization scales directly using data distribution statistics during each forward pass. Due to the fluctuating ranges in activations, these methods often result in training instability (Jacob et al., 2018). In quantization-aware training, follow-up methods (Esser et al., 2019; Bhalgat et al., 2020) propose novel strategies to learn quantization mapping in deep networks. Specifically, LSQ (Esser et al., 2019) approximates

the gradient to the quantizer step size (scaling factor), sensitive to quantized state transitions. In contrast, LSQ+ (Bhalgat et al., 2020) points out to involve learnable offsets for more effective asymmetric quantization. For super-resolution (SR) models (Lim et al., 2017), PAMS (Li et al., 2020) applies a trainable truncated parameter (can be regarded as quantization scale) to explore the upper bound of the activation quantization range to accommodate varying distributions of feature maps. Additionally, multi-bit quantization methods (Xu et al., 2022; Hong et al., 2022) provide another efficient multi-scale quantization approach, whereas there is limited hardware deployment support. We also notice that some Parameter-Efficient and Quantization-aware Adaptation (PEQA) methods (Kwon et al., 2022; Chen et al., 2024) introduce quantization to efficient fine-tuning of large language models (LLMs), where only the scales of weights are fine-tuned in specific task, enhancing the adaptability of the LLMs to downstream tasks. Unlike existing multi-bit quantization or PEQA, we identify the bottlenecks in co-training during QAT, focusing on the shared representation learning for quantized models with adaptive scales suitable for efficient lower-bit deployment.

## 3 PRELIMINARIES

### 3.1 CO-TRAINING FRAMEWORK

In a multi-task co-training, the architecture of models usually comprises multiple heads $\mathcal{H}_{\text{task}}(\cdot)$ and tails $\mathcal{T}_{\text{task}}(\cdot)$ to address each task independently, along with a shared body $\mathcal{B}(\cdot)$ and a task-specific query embedding $f_{\text{task}}$ (Hu & Singh, 2021; Likhosherstov et al., 2022; Girdhar et al., 2022; Srivastava & Sharma, 2024). During the training, the model is trained on multiple tasks. Still, for each batch, specific task input is selected based on a sampling strategy (Likhosherstov et al., 2022; Girdhar et al., 2022). The overall process is expressed as:

$$\hat{O} = \mathcal{T}_{\text{task}}\left(\mathcal{B}\left(\mathcal{H}_{\text{task}}(I), f_{\text{task}}\right)\right), \tag{1}$$

where $I$ denotes the input data for a specific task, and $\hat{O}$ represents the output generated by the network after processing through the head, shared body, and tail.

For instance, IPT (Chen et al., 2021), based on the degradation modeling of different low-level visual tasks, synthesizes various corrupted task-specific images from randomly selected data in large-scale ImageNet (Deng et al., 2009) dataset as task inputs for co-training. It achieves dominant high performances in super-resolution, deraining, and denoising.

### 3.2 QUANTIZATION BASELINES

We follow LSQ+ (Bhalgat et al., 2020) to introduce a general framework of symmetric channel-wise weight quantization and asymmetric layer-wise activation quantization. The quantization operations are defined as follows:

$$Q_a(x) = \left\lfloor clip\left\{\frac{x - z_x}{\alpha_x}, -Q_n^x, Q_p^x\right\}\right\rceil, \quad Q_w(w) = \left\lfloor clip\left\{\frac{w}{\alpha_w}, -Q_n^w, Q_p^w\right\}\right\rceil, \tag{2}$$

$$\hat{x} = Q_a(x) \circ \alpha_x + z_x, \quad \hat{w} = Q_w(w) \circ \alpha_w, \tag{3}$$

where $x$ denotes the activation value, $w$ denotes the weight value, and $z_x$ represents the zero-point offset for activation quantization. $\alpha_x$ and $\alpha_w$ are the scaling factors for activations and weights, respectively. With $a$-bit quantization, $Q_n^x = 2^{a-1}, Q_p^x = 2^{a-1} - 1$ are the discrete bounds. The function $clip\{y, r_1, r_2\}$ returns $y$ constrained between $r_1$ and $r_2$ (*i.e.*, values lower than $r_1$ are set to $r_1$, and values higher than $r_2$ are set to $r_2$). The operator $\lfloor y \rceil$ rounds $y$ to the nearest integer, and the $\circ$ denotes the channel-wise multiplication (layer-wise for activations with scalar $\alpha_x$). $\hat{x}$ and $\hat{w}$ represent the dequantized approximation values of the activations and weights, respectively. Denoting the loss function as $\mathcal{L}$, the straight-through estimator (STE) (Bengio et al., 2013) is used to retain the derivation of the gradient in backward propagation:

$$\frac{\partial \mathcal{L}}{\partial x} = \frac{\partial \mathcal{L}}{\partial \hat{x}}\frac{\partial \hat{x}}{\partial x} = \begin{cases} \frac{\partial \mathcal{L}}{\partial \hat{x}} & \text{if } x \in [-Q_n^x, Q_p^x] \\ 0 & \text{otherwise} \end{cases}. \tag{4}$$

We apply the quantization framework to perform 4-bit quantization on co-training models like IPT, establishing two baselines (**baseline-single** and **baseline-multi**). Given the full-precision pre-trained model, **baseline-single** uses the above quantization framework to quantize full-precision model for each task separately. Instead, **baseline-multi** co-trains the model with QAT directly. The details of **baseline-multi** and **baseline-single** are expatiated in Appendix A.

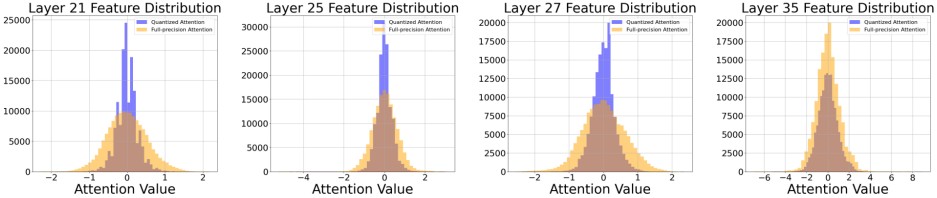

Figure 2: Comparison of PSNR on Set14 super-resolution benchmark for **baseline-multi** (red, BM), **baseline-single** (green, BS), and full precision IPT model (blue, FP).

Figure 3: Histogram of features produced by general attention calculation (orange) and quantized calculation (blue) from 4 selected attention layers in **baseline-multi** (4-bit quantized IPT model). We can observe that the distribution of quantized features differs noticeably from that of full-precision features, especially in the deeper layers, indicating an information distortion.

## 4 CHALLENGE ANALYSIS

We conduct and analyze the aforementioned baselines on IPT. Sec. 4.1 indicates that **baseline-multi** exhibits a more competitive performance. Sec. 4.2 points out the bottlenecks arising from the optimization competition among tasks, which can lead to mismatched quantization scales, affecting model performance.

### 4.1 CHOICE BETWEEN SINGLE-TASK AND MULTI-TASK QUANTIZATION

Considering that multi-task learning may lead to task interference problem (Maninis et al., 2019), a primary question is whether co-training with multiple tasks can be integrated with quantization while maintaining the superior performance of full-precision models. We compare the performance of **baseline-multi** and **baseline-single** on super-resolution tasks ($\times 2$, $\times 3$, $\times 4$ scale), as shown in Fig. 2. More results and evidence are presented in Sec. 6.1. Due to the regularization of co-training (Likhosherstov et al., 2022), the **baseline-multi** achieves better performance across multiple tasks compared to the **baseline-single**, which aligns with the training results of the full-precision model. Thus, it is worth noting that although the full-precision IPT exhibits prominent performance and generalization capabilities during co-training, the learned universal representation (Chen et al., 2021) does not translate well to a low-bit quantization counterpart when finetuning on a single task. Based on this observation, it is essential to incorporate quantization-aware training into co-training.

### 4.2 BOTTLENECK OF CO-TRAINING QUANTIZATION.

With the quantized co-training framework, we further discover that the **baseline-multi** still exhibits significant performance loss compared to the full-precision model, with varying gaps in performance across different tasks, as shown in Fig. 2. The analysis of the quantized feature distributions in the activation quantizers reveals that the incompatibility of existing QAT methods primarily stems from mismatched activation scales in the co-training framework. Specifically, we visualize the input feature distributions and their 0.995 / 0.005 quantiles for the activation quantizer of *decoder.layer9.self_attn.value* in **baseline-multi**, as shown in Fig. 1. It can be observed that, for the deraining task, the range of $[-3.72, 3.75]$ can cover 99% of its feature values, whereas denoising tasks require an interval of $[-7.66, 7.64]$. This indicates that different tasks reveal distinct data feature distributions, particularly with significant differences in the mean and variance between super-resolution and denoising tasks. Moreover, during training, the influence of different tasks on quantization scales may conflict, resulting in the failure of quantization for specific tasks. Conse-

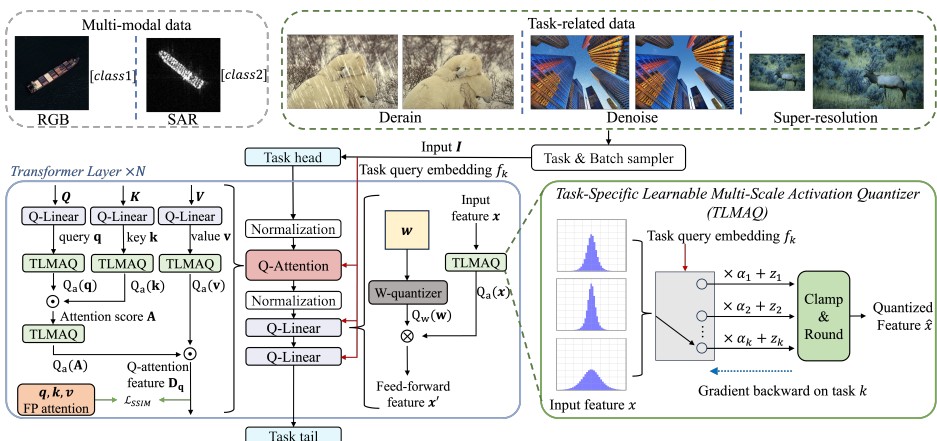

Figure 4: Overview of TSQ-MTC. Note that our TLMAQ can be employed for all activation quantization in both the linear layer and attention module of the Transformer, as well as the convolution layer in CNNs examined in our experiments.

quently, the scaling factors learned by the activation quantizer cannot effectively match all input data, negatively impacting the representational ability of the quantized features.

In addition, during co-training, we notice that quantization introduces information distortion and precision errors, which can be regarded as cumulative distribution perturbations (Li et al., 2022; Xu et al., 2023), especially within the attention module of Vision Transformers (Dosovitskiy, 2021). As shown in Fig. 3, there is a noticeable difference between the features produced by the general attention calculation and those generated by the quantized attention calculation, resulting in a loss of quantization performance.

## 5 THE PROPOSED METHOD

This section provides a detailed explanation of our Task-Specific Scales Quantization for Multi-Task Co-Training (TSQ-MTC), as shown in Fig. 4, effectively integrating quantization-aware training (QAT) with co-training. Our TSQ-MTC mainly consists of two novel techniques: task-specific learnable multi-scale activation quantizer (TLMAQ) and structure-based layer-by-layer distillation (SLLD), which are presented in Sec. 5.1 and Sec. 5.2, respectively.

### 5.1 ADAPTIVE QUANTIZATION SCALES FOR MULTI-TASK CO-TRAINING

Because LayerNorm (Vaswani, 2017) in Transformers and BatchNorm (Ioffe & Szegedy, 2015) in CNNs can not align the distributions of data in different tasks, multi-task learning may lead to degraded model performance, especially in quantization. The core of our approach is to design more suitable quantization mappings that allow the model to adapt simultaneously to the data distributions of different tasks. PAMS (Li et al., 2020) indicates that the range variations of the weights are stable. Therefore, adjusting the layer-wise activation quantizers is efficient and does not add overhead. We propose a task-specific learnable multi-scale activation quantizer (TLMAQ), which learns the corresponding activation quantizer scales for each task input to alleviate the quantization mapping conflicts between different tasks. Specifically, given $K$ training tasks, for the input data $x$ belonging to task $k$, there is a corresponding activation function:

$$Q_a(x, f_k) = \left\lfloor clip\left\{ \frac{x - \Phi_z(f_k)}{\Phi_\alpha(f_k)}, -Q_n^x, Q_p^x \right\} \right\rceil, \tag{5}$$

$$\hat{x} = Q_a(x, f_k) \circ \Phi_\alpha(f_k) + \Phi_z(f_k), \tag{6}$$

where $f_k$ is the query embedding of task $k$. $\Phi_\alpha(f_k)$ and $\Phi_z(f_k)$ are switching functions to select the learnable scaling factor $a_x^k$ and offset $z_x^k$ corresponding to task $k$ for activation quantizer respectively. Since the quantized feature is also inversely transformed to the original scale after completing the

Figure 5: Visualization of the task-related scaling factors for the activation quantizer in *encoder.layer1.self_attn.query*, *decoder.layer6.linear1*, *decoder.layer9.self_attn.value* of the IPT model trained with TLMAQ. It is noted that the scaling factors for the denoising task (red), deraining task (green), and super-resolution task (blue) show distinct differences.

low-bit calculation using Eq. 6, different quantization scales will not affect the floating-point range of the output feature. Our task-specific learnable multi-scale activation quantizer performs independent quantization for different task inputs, aligning with the co-training strategy and resulting in a simple and efficient implementation. Therefore, it can be used for the linear layer and attention module in the shared Transformer backbone and the convolution layer in CNNs, examined in Sec. 6. The overall process is summarized in Alg. 1.

**Initialization and convergence of scales.** Although the quantization scales of the model are learned through gradient updates, their initialization significantly impacts the network's convergence and performance (Bhalgat et al., 2020). Using the feature statistics of a pre-trained full-precision network as initialization enables the model to adapt effectively to training tasks (Li et al., 2020). Therefore, we initialize the weights from their full-precision counterparts at the start of training and then calculate the corresponding quantization scales $a_x^k$ and $z_x^k$ separately for each task $k$ based on the different inputs $x$ and task query embedding $f_k$. Experiments indicate that converged full-precision weight is indispensable for task-specific quantization scales; otherwise, TLMAQ may not lead to performance improvements.

**Task-scale consistency.** To validate the effectiveness of TLMAQ, we conduct a visual analysis of the quantization scales for the IPT model trained with our module. Among the various activation quantizers, the quantization scales exhibit significant task consistency, with similar tasks showing comparable quantization scales. In contrast, dissimilar tasks demonstrate considerable differences, as shown in Fig. 5. Specifically, the scales for $\times 2$, $\times 3$, and $\times 4$ super-resolution tasks are similar, whereas there is no apparent correlation of scales with the denoising tasks and super-resolution tasks. This observation aligns with the similarities in task distributions discussed in Sec. 4.2. Therefore, we suggest that quantization scales can be merged according to tasks after convergence, allowing similar tasks to share a set of quantization scales, further reducing the model's parameters.

## 5.2 STRUCTURE-BASED LAYER-BY-LAYER DISTILLATION

To mitigate the information distortion observed in Sec. 4.2, we adopt a layer-by-layer distillation following Wang et al. (2024). Specifically, in both the encoder and the decoder of the Transformer, **q**, **k**, and **v** simultaneously perform general and quantized attention calculation, yielding full-precision outputs $F$ and quantized outputs $\hat{F}$ for information alignment. However, vision tasks, especially low-level ones, are susceptible to information loss. Simple logit-level supervision is effective for classification tasks but coarse-grained and sub-optimal in this context. In contrast, structural similarity (SSIM) preserves spatial information more effectively, which is crucial for vision tasks (Xie et al., 2023). We compute the mean SSIM over the feature map using a sliding window. The quantized outputs $\hat{F}$ are refined with a distillation loss:

$$\mathcal{L}_{dis} = \lambda_{dis}(1 - \textbf{MSSIM}(F, \hat{F})), \tag{7}$$

where $\textbf{MSSIM}(\cdot)$ is the mean SSIM function and $\lambda_{dis}$ is the coefficient of distillation loss. This distillation method directly computes the full-precision attention results before quantization. Thus, it does not require a full-precision model, thereby reducing the resource requirements for training. For Transformer models, the final optimization objective is the combination of the task-specific loss and the SLLD loss. The effectiveness of SLLD can be found in Appendix D.

---

**Algorithm 1** Task-Specific Scales Quantization for Multi-Task Co-Training with TLMAQ

---

**Input:** Set of $K$ task-related training datasets $D = \{D_1, \ldots, D_k\}$, full-precision model $M_f$, task-specific activation quantizer scaling factor $\alpha_x^k$ and offset $z_x^k$
**Output:** Quantized model $M_q$
1: Initialize weights of $M_q$ from $M_f$ with Eq. 2;
2: **for** $i = 1$ to $K$ **do**
3:     Initialize $\alpha_x^k$ and $z_x^k$ with input from dataset $D_k$;
4: **end for**
5: **for** $i = 1$ to $N$ epoch **do**
6:     Randomly select a task $k$;
7:     Sample a mini-batch in $D_k$;
8:     Forward propagation with Eq. 5 and Eq. 6;
9:     Update weights and $\alpha_x^k$ and $z_x^k$ via gradient backward propagation;
10: **end for**
11: **return** $M_q$.

---

## 6   EXPERIMENTS

In this section, we evaluate the performance of TSQ-MTC in two settings within the co-training framework. The first experiment focuses on single-modal task-related data co-training, conducted on IPT, performing low-level vision tasks. The second experiment focuses on multi-modal data co-training and is conducted on shared-parameter CNN classification models. Comparisons with state-of-the-art single-task super-resolution quantization methods are provided in Appendix B. Additionally, we discuss the compression performance of TSQ-MTC compared to **baseline-multi**, including parameter counts and operations, in Appendix E.

### 6.1   SINGLE-MODAL DATA CO-TRAINING

**Task.** In this experiment, we focus on utilizing TSQ-MTC to improve the performance of a 4-bit quantized IPT model, which processes single-modal task-related image data. We compare our TSQ-MTC with **baseline-single** and **baseline-multi** based on LSQ+ (Bhalgat et al., 2020). In addition, we also show results of several quantization methods related to our work, including MinMax using statistics to calculate scales, LSQ (Esser et al., 2019) with learnable scales, PAMS (Li et al., 2020) designed for super-resolution and Q-ViT (Li et al., 2022) for Transformer-based model. Except **baseline-single**, all methods are trained in the co-training framework, including seven low-level vision tasks, such as super-resolution ($\times 2$, $\times 3$ and $\times 4$), deraining, and denoising ($\sigma = 30$ and $\sigma = 50$).

**Datasets.** We evaluate the super-resolution performace on Set5 (Bevilacqua et al., 2012), Set14 (Zeyde et al., 2012), B100 (Martin et al., 2001), Urban100 (Huang et al., 2015). For the denoising task, we adopt CBSD68 (Martin et al., 2001) and Urban100. As for deraining, we use Rain100L (Yang et al., 2017). For all benchmarks, the performances are measured by PSNR.

**Results.** Quantitative results for super-resolution tasks are shown in Table 1. The **baseline-single** achieves only 33.00 dB PSNR for the $\times 2$ scale super-resolution task on the Urban100 dataset, which is significantly lower than the 33.76 dB of the full-precision IPT. In contrast, the **baseline-multi** achieved 33.46 dB on the same task after co-training, demonstrating the effectiveness of co-training. Moreover, the PSNR on all super-resolution tasks achieved by our TSQ-MTC method is significantly higher than **baseline-multi**, showcasing the superior effectiveness of our adaptive scales. For example, the 4-bit IPT trained by our method achieves 27.08dB PSNR on Urban100 dataset for $\times 4$ SR, surpassing the **baseline-multi** by 0.23 dB. TSQ-MTC also attains higher PSNR than other existing methods, such as MinMax, LSQ, PAMS and Q-ViT. Furthermore, our approach reaches comparable results compared to full-precision counterpart on specific datasets ($\times 2$ super-resolution on Set14 / B100 and $\times 4$ super-resolution on Set5), while accelerates and compresses the model by $7.99\times$. Similar trends can also be found for denoising and deraining tasks in Appendix B. For instance, our TSQ-MTC reaches 42.02dB in deraining task on Rain100L dataset, while **baseline-multi** achieves only 41.89dB. The quantitative results show that our method surpasses other approaches, highlighting its overall performance.

Table 1: Comparison of methods across different super-resolution scales (PSNR(dB)). All methods are implemented on IPT for 4-bit quantization. $w$, $\alpha$ and $a$ represent the bit width of weights, activations, and attentions.

| Method | #Bits ($w/\alpha/a$) | Co-Training | Scale | Set 5 | Set 14 | B100 | Urban100 |
|---|---|---|---|---|---|---|---|
| Full Precision | 32/32/32 | ✓ | ×2 | 38.37 | 34.43 | 32.48 | 33.76 |
| baseline-single | | ✗ | | 38.13 | 34.03 | 32.32 | 33.00 |
| Minmax | | ✓ | | 37.16 | 32.78 | 31.62 | 30.64 |
| PAMS | | ✓ | | 38.12 | 34.14 | 32.35 | 33.12 |
| LSQ | 4/4/4 | ✓ | ×2 | 38.18 | 34.34 | 32.42 | 33.43 |
| baseline-multi | | ✓ | | 38.25 | 34.29 | 32.43 | 33.46 |
| Q-ViT | | ✓ | | 38.30 | 34.35 | 32.45 | 33.60 |
| **TSQ-MTC** | | ✓ | | **38.33** | **34.42** | **32.47** | **33.67** |
| Full Precision | 32/32/32 | ✓ | ×3 | 34.81 | 30.85 | 29.38 | 29.49 |
| baseline-single | | ✗ | | 34.58 | 30.59 | 29.23 | 28.91 |
| Minmax | | ✓ | | 33.44 | 29.61 | 28.59 | 26.92 |
| PAMS | | ✓ | | 34.52 | 30.59 | 29.22 | 28.91 |
| LSQ | 4/4/4 | ✓ | ×3 | 34.59 | 30.71 | 29.30 | 29.17 |
| baseline-multi | | ✓ | | 34.64 | 30.73 | 29.30 | 29.18 |
| Q-ViT | | ✓ | | 34.70 | 30.76 | 29.33 | 29.29 |
| **TSQ-MTC** | | ✓ | | **34.73** | **30.80** | **29.35** | **29.40** |
| Full Precision | 32/32/32 | ✓ | ×4 | 32.64 | 29.01 | 27.82 | 27.26 |
| baseline-single | | ✗ | | 32.37 | 28.79 | 27.67 | 26.76 |
| Minmax | | ✓ | | 31.17 | 27.88 | 27.10 | 25.06 |
| PAMS | | ✓ | | 32.31 | 28.76 | 27.65 | 26.67 |
| LSQ | 4/4/4 | ✓ | ×4 | 32.43 | 28.88 | 27.73 | 26.88 |
| baseline-multi | | ✓ | | 32.48 | 28.89 | 27.72 | 26.85 |
| Q-ViT | | ✓ | | 32.57 | 28.91 | 27.75 | 26.92 |
| **TSQ-MTC** | | ✓ | | **32.64** | **28.97** | **27.78** | **27.08** |

Table 2: Comparison of accuracy on different 4-bit quantized CNN models and modalities.

| Backbone | Modality | Full Precision Top-1 | LSQ+ Top-1 | TSQ-MTC Top-1 |
|---|---|---|---|---|
| ResNet-18 | All | 95.37 | 96.73 | **97.28** |
| | SAR | 94.54 | **96.17** | **96.17** |
| | RGB | 96.20 | 96.74 | **98.37** |
| ResNet-34 | All | 96.19 | 95.64 | **97.55** |
| | SAR | 93.99 | 92.90 | **96.17** |
| | RGB | 98.37 | 98.37 | **98.91** |
| ResNet-50 | All | 94.55 | 95.64 | **96.73** |
| | SAR | 91.80 | 92.35 | **93.44** |
| | RGB | 97.28 | 97.83 | **100.00** |
| ResNet-101 | All | 94.55 | 96.46 | **97.55** |
| | SAR | 91.80 | 93.44 | **96.72** |
| | RGB | 97.28 | **99.46** | 98.37 |

## 6.2 MULTI-MODAL DATA CO-TRAINING

**Tasks.** In this experiment, we focus on utilizing TSQ-MTC to enhance the performance of 4-bit quantized shared-parameter CNN classification models for multi-modal data. Our objective is to further validate the versatility and applicability of our TSQ-MTC. We consider the classification of different modalities as distinct tasks and modify our CNN backbones into multi-heads and multi-tails architectures. The implementation details are provided in Appendix A. The tasks involve classifying ships and airplanes based on SAR and RGB images. We compare our method with the general quantization framework in Sec. 3.2, namely LSQ+ (Bhalgat et al., 2020), to assess its effectiveness. Specifically, we use ResNet-18, ResNet-34, ResNet-50, and ResNet-101 (He et al., 2016) as shared-parameter backbones by adding heads and tails to process each modality. All models, including the full precision network, are trained within the co-training framework.

**Datasets.** For experiments on multi-modal data co-training, we utilize a SAR-RGB dataset collected from open-source datasets. Details of the dataset are provided in Appendix F.

Table 3: Ablation study on TLMAQ for super-resolution tasks. All methods are implemented on IPT for 4-bit quantization.

| Method | Scale | Set 5 | Set 14 | B100 | Urban100 |
|---|---|---|---|---|---|
| Baseline-multi | | 38.25 | 34.29 | 32.43 | 33.46 |
| w/ TLMAQ | ×2 | 38.32 | 34.38 | 32.46 | 33.59 |
| w/ TLMAQ & Init | | **38.34** | 34.34 | **32.47** | 33.64 |
| w/ TLMAQ & Init & SLLD (**TSQ-MTC**) | | 38.33 | **34.42** | 32.47 | **33.67** |
| Baseline-multi | | 34.64 | 30.73 | 29.30 | 29.18 |
| w/ TLMAQ | ×3 | 34.72 | 30.77 | 29.34 | 29.28 |
| w/ TLMAQ & Init | | 34.72 | **30.80** | **29.35** | 29.36 |
| w/ TLMAQ & Init & SLLD (**TSQ-MTC**) | | **34.73** | 30.80 | 29.35 | **29.40** |
| Baseline-multi | | 32.48 | 28.89 | 27.72 | 26.85 |
| w/ TLMAQ | ×4 | 32.60 | 28.94 | 27.76 | 26.95 |
| w/ TLMAQ & Init | | 32.61 | **28.97** | **27.78** | 27.06 |
| w/ TLMAQ & Init & SLLD (**TSQ-MTC**) | | **32.64** | **28.97** | **27.78** | **27.08** |

Table 4: Ablation study on SLLD. All methods are implemented on IPT for 4-bit quantization.

| Distillation Loss | None | Cross Entropy | Cosine Similarity | SSIM (TSQ-MTC) |
|---|---|---|---|---|
| **PSNR** | 37.596 | 37.698 | 37.713 | **37.893** |

**Results.** Our quantitative results, shown in Table 2, indicate that our method consistently outperforms LSQ+ across various backbones. Furthermore, our method performs better on deeper models, such as ResNet-34 (97.55% of all) and ResNet-50 (100% of RGB). Mismatching quantization scales may accumulate with deeper models, leading to significant performance degradation. Our approach effectively mitigates this issue, resulting in superior performance. It is worth noting that overfitting may occur when the network becomes too deep. As a result, the multi-task ResNet-101 trained by LSQ+ performs better in classifying RGB images but performs poorly on SAR images.

## 6.3 ABLATION STUDY

To demonstrate the effectiveness of the techniques in our TSQ-MTC framework, we conduct ablation studies on TLMAQ and SLLD.

**Effectiveness of TSQ-MTC.** For TLMAQ, ablation studies are conducted on single-modal task-related data co-training with 4-bit quantized IPT. We validate the effectiveness of TLMAQ, task-specific initialization, and SLLD. The resulting PSNR values of SR tasks are presented in Table 3. Results show that TLMAQ significantly improves performance compared to the **baseline-multi**, and the model achieves further improvement after incorporating the initialization strategy. Furthermore, with SLLD, our model exhibits noticeable improvements in specific metrics.

**Comparison of different distillation losses.** To further validate the effectiveness of SLLD, we use **baseline-single** for comparison. We train our models on DIV2K (Timofte et al., 2017) dataset in ×2 SR task for 150 epochs and evaluate the performance of different loss functions on Set5 benchmark, including cross-entropy, cosine similarity, and structural similarity. As shown in Table 4, the SLLD outperforms baseline and other loss functions, with 37.893dB PSNR, demonstrating the effectiveness of the structural constraints imposed on the attention module.

## 7 CONCLUSION

This paper proposes Task-Specific Scales Quantization for Multi-Task Co-Training (TSQ-MTC). This effective method addresses the bottleneck of combining co-training and quantization-aware training. Specifically, TSQ-MTC leverages a task-specific learnable multi-scale activation quantizer (TLMAQ) to enhance the representational ability of quantized features across different tasks and a structure-based layer-by-layer distillation (SLLD) technique to preserve the information from the full-precision features. Through comprehensive experiments, we demonstrate the effectiveness of TSQ-MTC and its components, TLMAQ and SLLD. Especially, we successfully achieved a 4-bit quantized low-level visual foundation model based on IPT, which outperforms existing state-of-the-art quantized super-resolution networks, demonstrating the superior performance of TSQ-MTC.

ACKNOWLEDGEMENTS

The work was supported by the National Key Research and Development Program of China (Grant No. 2023YFC3306401). This research was also supported by the Beijing Natural Science Foundation L244043 and L223024, the Zhejiang Provincial Natural Science Foundation of China under Grant LD24F020007, the National Natural Science Foundation of China under Grant 62076016, 62406298, and 623B2016, the "One Thousand Plan" projects in Jiangxi Province (Jxsq2023102268), and the Fundamental Research Funds for the Central Universities, Communication University of China (CUC24XT07).

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

# APPENDIX

## A  IMPLEMENTATION DETAILS

**Baseline-multi and baseline-single.** Both **baseline-multi** and **baseline-single** are quantized by LSQ+ (Bhalgat et al., 2020). The difference is that **baseline-multi** employs the co-training strategy used in IPT to learn a shared representation across different low-level vision tasks. On the ImageNet (Deng et al., 2009) dataset, **baseline-multi** is trained for 50 epochs. In contrast, **baseline-single** conducts single-task quantization training on the DIV2K (Timofte et al., 2017) dataset for each SR scale. Following Hong et al. (2022), **baseline-single** is trained for 900 epochs until convergence, achieving performance that is consistent with existing single-task super-resolution quantization models (Qin et al., 2024).

**TSQ-MTC on IPT.** TSQ-MTC is employed to quantize the IPT model for single-modal task-related data co-training. PyTorch is used to implement both our baselines and TSQ-MTC. The training is conducted on NVIDIA Tesla A100 GPUs, each with 80 GB memory, using the Adam optimizer with $\beta_1 = 0.9$ and $\beta_2 = 0.999$. We initialize all the quantized models from full-precision pre-trained weights and perform co-training for 50 epochs on the ImageNet (Deng et al., 2009) dataset. The learning rate starts at $5 \times 10^{-5}$ and decays to $2 \times 10^{-5}$ throughout the training, with a batch size of 225. SSIM is the loss for SLLD, with a weight coefficient of 0.01. To ensure a fair comparison with the full-precision IPT model, for each super-resolution task, we perform 30 epochs of single-task fine-tuning on the DIV2K dataset (Timofte et al., 2017) respectively, with a learning rate of 1e-6.

**TSQ-MTC on CNNs.** We use ResNet-18, ResNet-34, ResNet-50, and ResNet-101 as shared-parameter backbones for multi-modal data co-training. We add multiple task-specific heads (a convolutional block) before the backbone to process the input data. After the backbone, multiple task-specific fully connected layers are used for output classification. The training is conducted on 2 NVIDIA Tesla A5000 GPUs using the Adam optimizer, initialized with full-precision weights. We do co-training for 50 epochs on our SAR-RGB dataset with an initial learning rate of $1 \times 10^{-5}$ multiplied by 0.1 at epoch 25.

## B  ADDITIONAL EXPERIMENTAL RESULTS ON SINGAL-MODAL DATA CO-TRAINING

**Quantitative results.** As a multi-task model, the quantized IPT is also evaluated on denoising and deraining tasks. We compare our approach against Minmax, PAMS (Li et al., 2020), LSQ (Esser et al., 2019), Q-ViT (Li et al., 2022), and **baseline-multi**. Our method consistently demonstrates its superior performance, as shown in Table A for the denoising task and Table B for the deraining task. Specifically, the results show that the performance gap among different QAT methods is relatively small for the denoising and deraining tasks. For instance, in the denoising task ($\sigma = 30$), the difference of PSNR between TSQ-MTC and **baseline-multi** is only 0.01dB on CBSD68 dataset. This is due to the quantization scales being more suitable for these two tasks. Nevertheless, our TSQ-MTC still achieves the best performance.

**Qualitative results.** We visualize the 4x super-resolution task results on three selected images from the Urban100 (Huang et al., 2015) dataset in Fig. 2. It can be observed that TSQ-MTC better restores the fine details of the original image than other methods.

**Comparison with the state-of-the-art quantization methods for super-resolution.** We also compare our approach against several single-task quantized SR models, such as PAMS (Li et al., 2020), CADyQ (Hong et al., 2022) and QuantSR-T (Qin et al., 2024). Note that the backbone for PAMS and CADyQ is SRResNet (referred to as PAMS-C and CADyQ-C), while the backbone for QuantSR is SwinIR-S (referred to as QuantSR-T). We also add the results of the IPT quantized by PAMS on super-resolution tasks (referred to as PAMS-IPT). As shown in Table C, our method exhibits a clear performance advantage compared to single-task quantized super-resolution models. For example, for the $\times 2$ super-resolution task on the Urban100 benchmark, TSQ-MTC achieves a PSNR of 33.67dB, significantly surpassing the performance of QuantSR-T, which recorded 32.20dB.

Table A: Comparison of methods for denoising tasks (PSNR(dB)). All methods are implemented on IPT for 4-bit quantization. $w$, $\alpha$ and $a$ represent the bit width of weights, activations, and attentions.

| Method | #Bits ($w/\alpha/a$) | Task | CBSD68 | Urban100 |
|---|---|---|---|---|
| Full Precision | 4/4/4 | Denoising ($\sigma = 30$) | 32.40 | 34.10 |
| Minmax | 4/4/4 | Denoising ($\sigma = 30$) | 30.30 | 30.13 |
| PAMS | | | 32.37 | 33.92 |
| LSQ | | | 32.39 | 34.00 |
| baseline-multi | | | 32.39 | 33.99 |
| Q-ViT | | | **32.40** | 34.02 |
| **TSQ-MTC** | | | **32.40** | **34.05** |
| Full Precision | 4/4/4 | Denoising ($\sigma = 50$) | 29.99 | 31.67 |
| Minmax | 4/4/4 | Denoising ($\sigma = 50$) | 28.89 | 28.68 |
| PAMS | | | 29.97 | 31.47 |
| LSQ | | | 29.97 | 31.56 |
| baseline-multi | | | 29.98 | 31.54 |
| Q-ViT | | | 29.98 | 31.59 |
| **TSQ-MTC** | | | **29.99** | **31.61** |

Table B: Comparison of methods for deraining task (PSNR(dB)). All methods are implemented on IPT for 4-bit quantization. $w$, $\alpha$, and $a$ represent the bit width of weights, activations, and attentions.

| Method | #Bits ($w/\alpha/a$) | Rain100L |
|---|---|---|
| Full Precision | 4/4/4 | 41.98 |
| Minmax | 4/4/4 | 35.91 |
| PAMS | | 41.46 |
| LSQ | | 41.87 |
| baseline-multi | | 41.89 |
| Q-ViT | | 41.98 |
| **TSQ-MTC** | | **42.02** |

Table C: Comparison of methods across ×2 and ×4 super-resolution scales (PSNR(dB)). $w$, $\alpha$ and $a$ represent the bit width of weights, activations, and attentions.

| Method | Backbone | Training | #Bits ($w/\alpha/a$) | Scale | Set 5 | Set 14 | B100 | Urban100 |
|---|---|---|---|---|---|---|---|---|
| PAMS-C | SRResNet | single-task | 4/4/4 | ×2 | 37.67 | 33.19 | 31.90 | 31.10 |
| CADyQ-C | SRResNet | single-task | | | 37.58 | 33.14 | 31.87 | 30.94 |
| QuantSR-T | SwinIR | single-task | | | 38.10 | 33.65 | 32.21 | 32.20 |
| PAMS-IPT | IPT | multi-task | | | 38.12 | 34.14 | 32.35 | 33.12 |
| TSQ-MTC | IPT | multi-task | | | **38.33** | **34.42** | **32.47** | **33.67** |
| PAMS-C | SRResNet | single-task | 4/4/4 | ×4 | 31.59 | 28.20 | 27.32 | 25.32 |
| CADyQ-C | SRResNet | single-task | | | 31.48 | 28.05 | 27.21 | 25.09 |
| QuantSR-T | SwinIR | single-task | | | 32.18 | 28.63 | 27.59 | 26.11 |
| PAMS-IPT | IPT | multi-task | | | 32.31 | 28.76 | 27.65 | 26.67 |
| TSQ-MTC | IPT | multi-task | | | **32.64** | **28.97** | **27.78** | **27.08** |

Table D: Ablation study on TLMAQ for denoising tasks (PSNR(dB)). All methods are implemented on IPT for 4-bit quantization.

| Method | Task | CBSD68 | Urban100 |
|---|---|---|---|
| Baseline-multi | | 32.39 | 33.99 |
| w/ TLMAQ | Denoising ($\sigma = 30$) | **32.40** | 34.02 |
| w/ TLMAQ & Init | | **32.40** | 34.04 |
| w/ TLMAQ & Init & SLLD (**TSQ-MTC**) | | **32.40** | **34.05** |
| Baseline-multi | | 29.98 | 31.54 |
| w/ TLMAQ | Denoising ($\sigma = 50$) | **29.99** | 31.57 |
| w/ TLMAQ & Init | | **29.99** | 31.59 |
| w/ TLMAQ & Init & SLLD (**TSQ-MTC**) | | **29.99** | **31.61** |

Table E: Ablation study on TLMAQ for deraining task (PSNR(dB)). All methods are implemented on IPT for 4-bit quantization.

| Method | Task | Rain100L |
|---|---|---|
| Baseline-multi | | 41.89 |
| w/ TLMAQ | Deraining | 41.99 |
| w/ TLMAQ & Init | | 41.98 |
| w/ TLMAQ & Init & SLLD (**TSQ-MTC**) | | **42.02** |

## C  ABLATION STUDY ON TLMAQ FOR DENOISING AND DERAINING

Ablation experiments are also performed on the denoising task and deraining task with 4-bit quantized IPT within the single-modal task-related data co-training setup. We evaluate the effectiveness of TLMAQ, task-specific initialization, and SLLD. The results of the denoising task are shown in Table D, while the results of deraining task are presented in Table E. We can see that TLMAQ improves performance over the **baseline-multi**, with more gains achieved through task-specific initialization. The addition of SLLD provides a further boost, resulting in the best performance across both tasks.

## D  EFFECTIVENESS OF SLLD

We visualize the distributions of full-precision and quantized attention results in the 4-bit quantized IPT, comparing models trained with and without SLLD, as shown in Fig. B. It can be observed that after adding SLLD, the distribution of quantized features aligns more closely with that of full-precision features, demonstrating the effectiveness of structural loss in preserving full-precision feature information.

## E  COMPRESSION RATIO

In Table F, we present the parameter counts and operations (*i.e.*, Ops) of the **baseline-multi** and the IPT trained by TSQ-MTC, as well as the full-precision model, along with their PSNR performances on the Urban100 benchmark for ×2 super-resolution task. For the super-resolution, we do not quantize the input and output layers of the model. Additionally, the upsampling layer at the tail's end is not quantized. It can be observed that **baseline-multi** and TSQ-MTC successfully achieve significant acceleration, with a compression ratio of 87.5%. Notably, TSQ-MTC achieves better performance by introducing a small number of additional activation quantizer scales, improving the PSNR from 33.46dB to 33.67dB, further narrowing the gap with the full-precision model. In fact, the parameter count of TSQ-MTC is almost identical to that of the **baseline-multi**, with merely an increase of 0.04M, demonstrating the efficiency of our method.

## F  DETAILS OF SAR-RGB DATASET

Due to the fact that existing classification datasets primarily contain a single modality and considering the significant quality differences between SAR and RGB datasets, we organize a dataset suitable

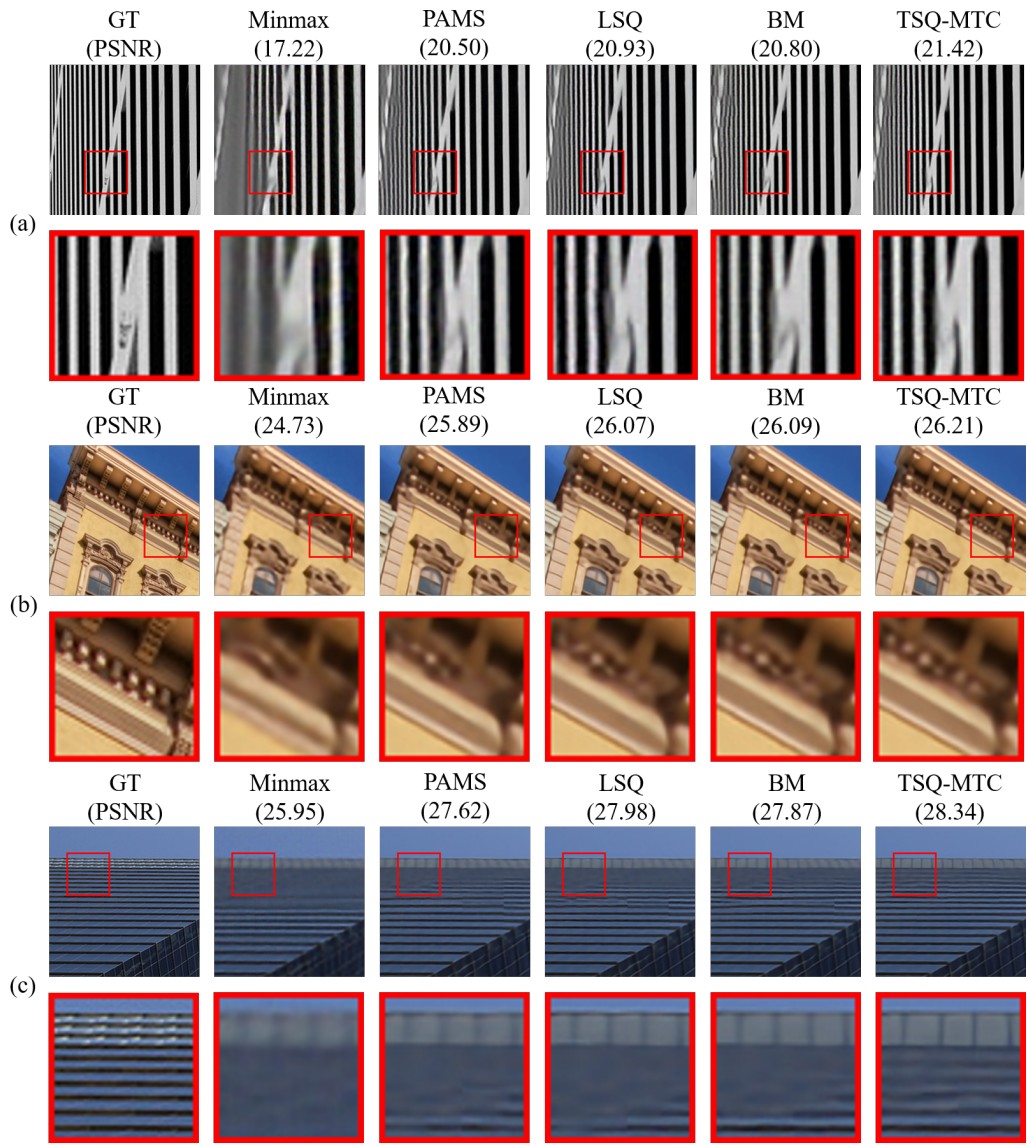

Figure A: Visualization of results for 4x super-resolution task (on selected images of Urban100).

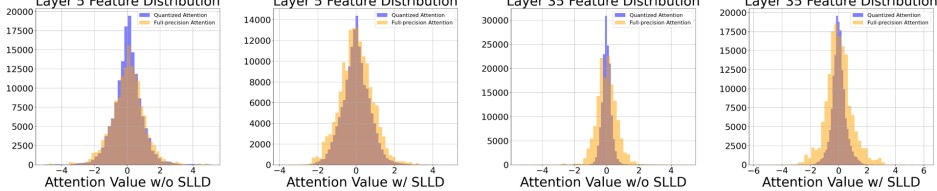

Figure B: Histogram of features produced by general attention calculation (orange) and quantized calculation (blue) from 4 selected attention layers in models with and without SLLD. It can be seen that SLLD can reduce the information distortion caused by quantization.

for the tasks described in this paper from open-source datasets. We manually segment and annotate these images, ultimately creating a high-quality dataset with aligned categories and consistent quantities across the two modalities. Some sample instances are illustrated in Fig. C. Specifically, our

Table F: Efficiency comparison in terms of Bit Width, Parameter Count, Ops, and PSNR for $\times 2$ super-resolution on Urban100 (Huang et al., 2015). The input image has a shape of $(1, 3, 48, 48)$. $w$ and $a$ represent the bit width of weights and activations.

| Method | #Bits ($\omega/\alpha/a$) | Params (M) ($\downarrow$Ratio) | Ops (G) ($\downarrow$Ratio) | Set14 PSNR | Urban100 PSNR |
|---|---|---|---|---|---|
| Full Precision | 32/32/32 | 115.31 ($\downarrow$0.0%) | 33.06 ($\downarrow$0.0%) | 34.43 | 33.76 |
| baseline-multi | 4/4/4 | 14.34 ($\downarrow$87.6%) | 4.14 ($\downarrow$87.5%) | 34.29 | 33.46 |
| TSQ-MTC | 4/4/4 | 14.38 ($\downarrow$87.5%) | | **34.42** | **33.67** |

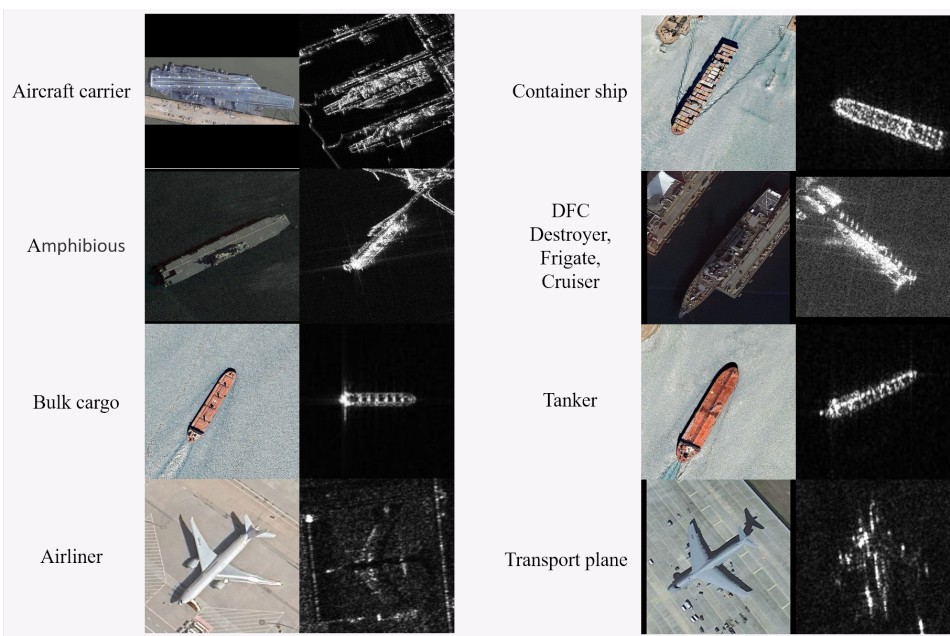

Figure C: Visualization of the SAR-RGB dataset.

SAR-RGB dataset is sourced from several open source datasets (Hou et al., 2020; Huang et al., 2017; Di et al., 2021; Xia et al., 2018), collected from Google Earth (satellite), TerraSAR, RadarSat, Capella, Gaofen-3, consisting of 1709 instances within 8 RGB categories and corresponding 8 SAR categories.

# G ACCURACY DURING TRAINING OF MULTI-TASK CNN MODEL

We visualize the validation accuracy during the training of the multi-task ResNet-34 quantized with TSQ-MTC and LSQ+, as shown in Fig. D. It is evident that in the later stage of training, our method significantly outperforms both LSQ+ and the full-precision model, while the models trained with LSQ+ underperform compared to the full-precision model for a period of time.

It is worth noting that the convergence of the full-precision model directly impacts the quantization performance of TSQ-MTC. Specifically, we utilize TSQ-MTC to quantize a ResNet model trained for only 30 epochs in full precision, while the models employed in our main experiments are trained for 50 epochs. The accuracy of the model after quantization is not significantly different from that of the same model quantized by LSQ+.

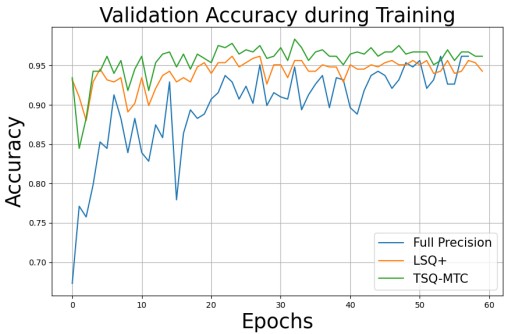

Figure D: Validation accuracy during training of quantized multi-task ResNet-34.

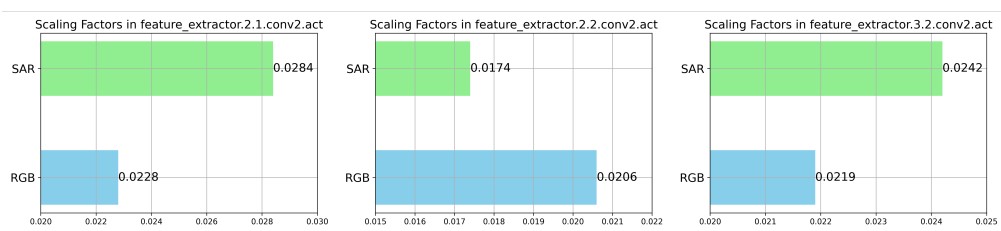

Figure E: Visualization of the task-related scaling factors for the activation quantizer in *feature_extractor.2.1.conv2.act*, *feature_extractor.2.2.conv2.act*, *feature_extractor.3.2.conv2.act* of quantized multi-task ResNet-34. It is noted that the scaling factors for SAR (green) and RGB (blue) show distinct differences.

# H   VISUALIZATION OF THE TASK-RELATED SCALING FACTORS OF MULTI-TASK CNN MODEL

We conduct a visual analysis of the quantization scales for the CNN model trained with TSQ-MTC. The scaling factors of the three modules in quantized multi-task ResNet-34 network, *feature_extractor.2.1.conv2.act*, *feature_extractor.2.2.conv2.act*, and *feature_extractor.3.2.conv2.act*, are visualized in Fig. E. We also observed pronounced differences in scales between different tasks, further supporting the validity of our approach. For example, in *feature_extractor.2.1.conv2.act*, the scaling factor for the SAR task is 1.23 times that of the RGB task, which can lead to significant changes in the quantization mapping.

# I   EXPERIMENTAL RESULTS ON HIGH-LEVEL TASKS

**Task and dataset.** To verify the effectiveness of TSQ-MTC in more scenarios, we conduct experiments on high-level vision tasks. We train a multi-task model on the NYUD-V2 (Silberman et al., 2012) dataset based on the code of DiSparse (Sun et al., 2022) within co-training. The tasks include semantic segmentation for RGB input and surface normal prediction for RGBD input. Each task is assigned a corresponding input head and output tail, while sharing a ResNet-based backbone. We first train a converged full-precision model to initialize the 4-bit quantized model. Our evaluation compares TSQ-MTC with **baseline-multi** (LSQ+) using four evaluation metrics: mIoU, Pixel Accuracy (Pixel Acc), Angle Mean Error (Mean Err) and Angle Median Error (Median Err).

**Results.** The results in high-level vision tasks are shown in Table G. It can be observed that our method significantly improves the performance of the semantic segmentation task (mIoU increased by 0.25%) while maintaining the performance of the surface normal prediction task. This demonstrates that our method effectively alleviates quantization conflicts between tasks.

Table G: Comparison of methods on high-level vision tasks for 4-bit quantization.

| Method | Semantic Segmentation | | Surface Normal Prediction | |
|---|---|---|---|---|
| | mIoU ↑ | Pixel Acc ↑ | Mean Err ↓ | Median Err ↓ |
| Full Precision | 23.15 | 55.18 | 17.05 | 15.85 |
| baseline-multi | 23.22 | 55.16 | **16.97** | 15.17 |
| TSQ-MTC | **23.47** | **55.27** | **16.97** | **15.16** |

## J  EXPERIMENTAL RESULTS ON CROSS-DOMAIN DATASET

**Task and dataset.** We also conduct experiments to explore the effectiveness of TSQ-MTC on cross-domain datasets. Specifically, we select a subset of DomainNet (Peng et al., 2019) for cross-domain classification tasks, which includes 13504 instances within four domains and ten categories for each domain. Consistent with the experiments in the main paper, we follow the DomainNet to use ResNet-101 (He et al., 2016) as the backbone and consider the classification of the four domains as four distinct tasks, adding corresponding heads and tails to process each modality. We compare TSQ-MTC with LSQ+ (Bhalgat et al., 2020).

**Results.** Experimental results show that compared to the full-precision model, both TSQ-MTC and LSQ+ achieved effective quantization. However, our method does not exhibit significant advantages. To investigate this finding, we further examine the quantization scales for different tasks within the network and found that the quantization scales of the same activation quantizer are almost the same. One possible explanation is that for cross-domain data within the same task, the feature distribution differences are negligible, and different tasks do not exert opposing effects on the scales. Using a single scale might actually lead to faster convergence.

