# OpenReview forum: "Efficient Low-Bit Quantization with Adaptive Scales for Multi-Task Co-Training"
_ICLR.cc/2025/Conference — ICLR 2025 Poster_

### Official Review · Reviewer_icGh · 2024-11-01

**Soundness:** 3
**Presentation:** 3
**Contribution:** 3
**Rating:** 8
**Confidence:** 5

**Summary:**

This paper presents a novel method for multi-task network quantization-aware training, addressing performance degradation caused by model quantization. The approach incorporates two techniques: (1) Task-Specific Learnable Multi-Scale Activation Quantizer (TLMAQ): Addressing the scale conflict in quantizing diverse task features, improving the quantized model's representational capabilities across tasks. (2) Structure-based Layer-by-Layer Distillation (SLLD): Strengthening full-precision model supervision over quantized models, reducing information distortion from quantization.

**Strengths:**

(1) The proposed method effectively addresses the scale discrepancies among multi-task features, aligning with intuitive expectations.
(2) The novelty is good, the introduction of task-aware quantization into multi-task learning represents an innovative approach.
(3) Method is architecture-agnostic, featuring a wide range of applicability.
(4) originality, quality, clarity, and significance: This paper demonstrates good originality, is well-written, and clearly communicates ideas. Furthermore, the application of quantization to enhance model efficiency holds significant importance for multi-task learning

**Weaknesses:**

(1) There is a minor error in Section 2.1 - regarding the order of description for the "single-modal task-related data scenario" and the "multi-modal data scenario" are misaligned with the order of "former scenario" and "latter scenario".
(2) The paper only considered the low-level multi-task scenarios; however, However, it lacks effective exploration of high-level visual tasks. Specifically, Can the proposed method still maintain effectiveness when processing high-level visual multitasking data such as NYUD-v2, Pascal Context and Cityscapes.
(3) The experiment involved a comparison of model quantization techniques, including LSQ (2019) and PAMS (2020). Nevertheless, a comparison with the most recent Quantization-Aware Training (QAT) methods is conspicuously absent.

**Questions:**

Apart from the points (2) and (3) outlined in the "Weaknesses" section, the following are additional concerns:
(1) While the model design is good and interesting, the model seems a little complex. Would it be possible to provide some comparative analyses regarding the model's speed and complexity?

---

> ### Author Response · Authors · 2024-11-21
> **Response to Reviewer icGh (Part 1)**
>
> **Thank you for your patience, as the additional experiments required more time.** \
> **We sincerely appreciate Reviewer icGh’s valuable feedback and constructive comments. We address each question below.**
>
> **Q1 (Weaknesses1):**
> There is a minor error in Section 2.1 - regarding the order of description for the "single-modal task-related data scenario" and the "multi-modal data scenario" are misaligned with the order of "former scenario" and "latter scenario".
>
> **A1:**
> We apologize for the mistake and thank you for pointing it out. We will correct it in the revised manuscript.
>
> **Q2 (Weaknesses2):**
> The paper only considered the low-level multi-task scenarios; however, However, it lacks effective exploration of high-level visual tasks. Specifically, Can the proposed method still maintain effectiveness when processing high-level visual multitasking data such as NYUD-v2, Pascal Context and Cityscapes.
>
> **A2:**
> Thank you for raising this question. TSQ-MTC is primarily designed for co-training and, for the first time, we introduce an effective quantization approach tailored for this framework. For validation, we select low-level vision tasks (IPT) and high-level classification task (ResNet), which are suitable for co-training and practical to implement. As you suggested, we share a strong interest in verifying the effectiveness of our method on more high-level vision tasks. However, these tasks often involve **resource-demanding** pre-training, such as unified architectures [1] and multi-modal large models [2]. Due to the resource and time constraints during the rebuttal period, it may not be feasible to fully explore these tasks at this stage. Therefore, the distribution differences of features in distinct high-level visual tasks require further investigation. According to our present observations, the distribution differences are likely to be widespread and may also occur in high-level vision tasks, for which our method can provide an effective solution. We are currently designing a validation experiment, but we may not be able to reach a conclusion within the rebuttal period. If we are able to complete the experiment during the rebuttal period, we will share the results with you. Otherwise, we will discuss the possibility of extending our work to higher-level tasks in the future.
>
> **Q3 (Weaknesses4):**
> The experiment involved a comparison of model quantization techniques, including LSQ (2019) and PAMS (2020). Nevertheless, a comparison with the most recent Quantization-Aware Training (QAT) methods is conspicuously absent.
>
> **A3:**
> Thank you for your valuable suggestion. The QAT methods compared in the main paper primarily relate to the quantization scales we focus on. As suggested, we include Q-ViT [3] as a new comparison method, which addresses the distribution degradation in quantized ViT. Q-ViT is a recent, highly-cited, and effective QAT method, which can serve as a representative comparative approach. The experimental results are presented in the tables below.
> | Method   | Task          | Set5     | Set14    | B100     | Urban100 |
> |----------|---------------|----------|----------|----------|----------|
> | Q-ViT    | $\times 2$ SR | 38.30    | 34.35    | 32.45    | 33.60    |
> | TSQ-MTC  | $\times 2$ SR | **38.33**| **34.42**| **32.47**| **33.67**|
>
> | Method   | Task          | Set5     | Set14    | B100     | Urban100 |
> |----------|---------------|----------|----------|----------|----------|
> | Q-ViT    | $\times 3$ SR | 34.70    | 30.76    | 29.33    | 29.29    |
> | TSQ-MTC  | $\times 3$ SR | **34.73**| **30.80**| **29.35**| **29.40**|
>
> | Method   | Task          | Set5     | Set14    | B100     | Urban100 |
> |----------|---------------|----------|----------|----------|----------|
> | Q-ViT    | $\times 4$ SR | 32.57    | 28.91    | 27.75    | 26.92    |
> | TSQ-MTC  | $\times 4$ SR | **32.64**| **28.97**| **27.78**| **27.08**|
>
> | Method   | Task          | CBSD68   | Urban100 |
> |----------|---------------|----------|----------|
> | Q-ViT    | Denoising 30  | **32.40**| 34.02    |
> | TSQ-MTC  | Denoising 30  | **32.40**| **34.05**|
>
> | Method   | Task          | CBSD68   | Urban100 |
> |----------|---------------|----------|----------|
> | Q-ViT    | Denoising 50  | 29.98    | 31.59    |
> | TSQ-MTC  | Denoising 50  | **29.99**| **31.61**|
>
> | Method   | Task          | Rain100L |
> |----------|---------------|----------|
> | Q-ViT    | Deraining     | 41.98    |
> | TSQ-MTC  | Deraining     | **42.02**|
>
> As shown in the tables above, Q-ViT, as a popular and recent QAT method, achieves significant performance improvements on Transformer-based models. However, the performance of our method still surpasses that of Q-ViT (e.g. +0.16dB on Urban100 for $\times 4$ SR). Notably, the motivation and implementation of our method differ from those of Q-ViT, and combining the two approaches could further enhance the performance of quantized models. \
> We will include these experimental results in the revised paper.

---

> ### Author Response · Authors · 2024-11-21
> **Response to Reviewer icGh (Part 2)**
>
> **Q4 (Questions1):**
> While the model design is good and interesting, the model seems a little complex. Would it be possible to provide some comparative analyses regarding the model's speed and complexity?
>
> **A4:**
> Thank you for your concern. Compared to existing QAT methods, our method does not add much complexity. The comparative analyses are discussed in Appendix Section E. We provide explanations and comparisons from two perspectives: computation cost and parameter counts. \
> (1) As the co-training [1, 2] framework trains a versatile model that performs only one task at a time for a given input, only the corresponding set of quantization scales is used during inference for each task, thereby avoiding any additional computational burden. As shown in Table F, compared to **baseline_multi** (LSQ+), the model trained with TSQ-MTC achieves the same 4.14G FLOPs with a compression ratio of  87.5%, while delivering a significant performance improvement (+0.21dB on the
> Urban100 dataset for $\times 2$ SR task). \
> (2) Since most low-bit quantization methods use layer-wise quantization for activations [4], we only chose to introduce task-specific quantization scales for the activation quantizer to adapt to different input data. The additional parameter count is minimal. Table F indicates that TSQ-MTC maintains an almost identical parameter count (14.38M with a compression ratio of 87.5% vs. 14.34M with 87.6%), with only a minor increase of 0.04M. \
> Additionally, SLLD is only adopted during training. As a result, our model is both effective and efficient.
>
> **Thank you once again for the insightful feedback. If you have any other questions or would like to discuss further, please inform us. We sincerely look forward to your comment.**
>
> **REFERENCES:**
>
> [1] Omnivore: A single model for many visual modalities. CVPR'2022. \
> [2] Minigpt-v2: large language model as a unified interface for vision-language multi-task learning. arXiv'2023 (395 citations). \
> [3] Q-vit: Accurate and fully quantized low-bit vision transformer. NeurIPS'2022 (77 citations). \
> [4] Cadyq: Content-aware dynamic quantization for image super-resolution. ECCV'2022.

---

> ### Author Response · Authors · 2024-11-27
> **High-Level Experiments**
>
> **For your concern about the performance of our method on high-level vision tasks, we design and conduct a validation experiment during the rebuttal period.**
>
> As you suggested, we train a multi-task model on the NYUD-V2 [1] dataset based on the code of DiSparse [2] within co-training. The tasks includes semantic segmentation (seg) for RGB input and surface normal prediction (sn) for RGBD input. Each task is assigned a corresponding input head and output tail, while sharing a ResNet-based backbone. We first train a converged full-precision model to initialize the 4-bit quantized model. Our experiment compares TSQ-MTC with **baseline-multi** (LSQ+) with four evaluation metrics. Due to time constraints, our full-precision model may not be trained with the optimal parameters. The results are shown in the table below.
>
> |Method            |seg-mIoU $\uparrow$|seg-Pixel Acc $\uparrow$|sn-Angle Mean Err $\downarrow$|sn-Angle Median  Err $\downarrow$|
> |------------------|---------|-------------|--------------|---------|
> |Full-Precision                |23.15    |55.18        |17.05         |15.85    |
> |Baseline-multi    |23.22    |55.16        |**16.97**     |15.17    |
> |TSQ-MTC           |**23.47**|**55.27**    |**16.97**     |**15.16**|
>
> It can be observed that our method significantly improves the performance of the semantic segmentation task (mIoU increased by 0.25%) while maintaining the performance of the surface normal prediction task. This demonstrates that our method effectively alleviates quantization conflicts between tasks.
>
> **We hope the above experiments address your concern. We will continue to complete the experiments on high-level vision tasks and include the results in the final version of the paper.**
>
> **REFERENCES:**
>
> [1] Indoor segmentation and support inference from rgbd images. ECCV'2012 \
> [2] DiSparse: Disentangled Sparsification for Multitask Model Compression. CVPR'2022.

---

### Official Review · Reviewer_gsJW · 2024-11-02

**Soundness:** 3
**Presentation:** 4
**Contribution:** 3
**Rating:** 6
**Confidence:** 3

**Summary:**

This paper proposes Task-Specific Scales Quantization for Multi-Task Co-Training (TSQ-MTC) to address the performance degradation issue of existing quantization-aware training (QAT) methods when integrated with co-training. The proposed method introduces a task-specific learnable multi-scale activation quantizer (TLMAQ) to enrich the representational ability of shared features across different tasks and a structure-based layer-by-layer distillation (SLLD) to ensure that the quantized features effectively preserve the information from their full-precision counterparts. Extensive experiments on two co-training data scenarios demonstrate the effectiveness of TSQ-MTC, which achieves a 4-bit quantized low-level visual foundation model based on IPT with a PSNR comparable to the full-precision model and a 7.99× compression ratio in the ×4 super-resolution task on the Set5 benchmark.

**Strengths:**

1. The authors provide a comprehensive evaluation of the challenges of the task, i.e., directly integrating multi-task co-training with QAT. This helps clarify the bottleneck of existing QAT methods and motivates the proposed TSQ-MTC method to address the performance degradation issue.

2. The proposed TSQ-MTC method introduces two novel components, TLMAQ and SLLD, to enhance the representational ability of shared features across different tasks and preserve the information from full-precision features in deeper layers of the Transformer model.

3. The experimental evaluation across the main text and appendices provides detailed insights into the effectiveness and versatility of the proposed TSQ-MTC method in two co-training data scenarios.

**Weaknesses:**

1. This paper referred the challenge of computational and memory overhead of co-trained models (Lines 047-049), but the computational complexity and efficiency of the proposed TSQ-MTC method are not discussed in detail.

2. The authors detailedly analysis the proposed SLLD in Table 4 and Figure B. However, according to the ablation studies provided in Table 3, Table D, Table E, the proposed SLLD has minor improvements in performance given the potential for minor fluctuations in experimental results.

**Questions:**

As listed in Weaknesses, there are two questions:
1. Could the authors provide the computational cost of training and inference compared to existing QAT methods?
2. Could the authors report the variance of the results of the proposed SLLD method in the ablation studies?

---

> ### Author Response · Authors · 2024-11-21
> **Response to Reviewer gsJW**
>
> **Thank you for your patience, as the additional experiments required more time.** \
> **We sincerely appreciate Reviewer gsJW’s valuable feedback and constructive comments. We address each question below.**
>
> **Q1:**
> Could the authors provide the computational cost of training and inference compared to existing QAT methods?
>
> **A1:**
> Thank you for your question.
> Our method does not significantly increase the computational cost. The relevant discussion, including FLOPs and parameter count, are provided in Appendix Section E. We provide some additional details for both training and inference. \
> (1) During training, the layer-wise activation quantization scales (scalars) do not occupy excessive GPU memory. Additionally, the model performs only one task per input batch with task-specific quantization scales. Therefore, TLMAQ does not increase the computational burden. For example, as shown in the table below, the 4-bit IPT quantized by TSQ-MTC uses 76,722 MiB of GPU memory with a batchsize of 45, whereas Q-ViT [1], a popular and recent QAT method also based on LSQ+, requires 78,996 MiB. The training times for both methods are roughly the same, taking about 4-5 days (on 5 NVIDIA A100 GPUs).
>
> |   Method  | Training GPU Memory | Training Time   |
> |-----------|---------------------|-----------------|
> | TSQ-MTC   | 76,722 MiB          | $\sim$ 4.5 days |
> | Q-ViT     | 78,996 MiB          | $\sim$ 5 days   |
>
> (2) For inference, we compare the parameter counts and FLOPs of the 4-bit IPT trained by **baseline-multi** (LSQ+) and TSQ-MTC in Table F. Our TSQ-MTC outperforms **baseline_multi** while maintaining a nearly identical parameter count (14.38M with a compression ratio of 87.5% vs. 14.34M with 87.6%), with only a slight increase of 0.04M. Moreover, our method has the same FLOPs (4.14G) and acceleration performance as LSQ+. \
> We will emphasize the computational efficiency of TSQ-MTC in the revised paper.
>
> **Q2:**
> Could the authors report the variance of the results of the proposed SLLD method in the ablation studies?
>
> **A2:**
> Thank you for your suggestion. Regarding your concern, we conduct our discussion by analyzing performance observations and providing further validations. \
> (1) **Performance Observations**: In our experiments, we find that improving the performance of quantized IPT on certain datasets can be challenging, such as the denoising task on the CBSD68 dataset (Table A). SLLD, as an additional enhancement to our method, is primarily aimed at addressing the observed information distortion issue (Figure B). Therefore, the improvement is less pronounced than that of TLMAQ. Addationally, after incorporating SLLD, the performance consistently improves or remains stable. It also demonstrates clear performance improvements on certain complex datasets, such as Urban100 (+0.03dB for $\times 3$ SR). \
> (2) **Further Validations**: To address your concern about potential experimental fluctuations, we conduct additional experiments to obtain the variance. Given the limited time during the rebuttal period, we resume training from the weights at the 40th epoch and calculate the mean and variance of the test results for the final weights (50 epochs) of SLLD and non-SLLD training. Each set of statistics includes five test results on the same task and dataset. The calculated standard deviation results are as follows:
> | Method                       | $\times 2$ SR on Set 14       | $\times 3$ SR on Set 5        | $\times 4$ SR on Set 5        |
> |------------------------------|------------------|----------------|-----------------|
> | w/ TLMAQ & Init              | 34.337 ± 0.005   | 34.691 ± 0.011   | 32.576 ± 0.011   |
> | w/ TLMAQ & Init & SLLD       | **34.404 ± 0.006** | **34.711 ± 0.007** | **32.613 ± 0.013** |
>
> It can be seen that the standard deviation of our method is not significant enough to affect its performance.
> We hope the table above clarifies your concerns and provides further evidence of the robustness of our approach.
>
> **Thank you once again for the insightful feedback. If you have any other questions or would like to discuss further, please inform us. We sincerely look forward to your comment.**
>
> **REFERENCES:**
>
> [1] Q-vit: Accurate and fully quantized low-bit vision transformer. NeurIPS'2022.

---

> > ### Comment · Reviewer_gsJW · 2024-11-27
> >
> > Thanks for your time and efforts in additional experiments, which address my concerns.

---

> > > ### Author Response · Authors · 2024-11-27
> > >
> > > We sincerely appreciate your suggestions and feedback. We are glad to have addressed all of your concerns and hope that you will consider raising our score.

---

### Official Review · Reviewer_LoRG · 2024-11-03

**Soundness:** 4
**Presentation:** 4
**Contribution:** 3
**Rating:** 6
**Confidence:** 3

**Summary:**

The study finds that directly applying co-training to existing QAT methods significantly degrades performance. The main issue identified is the inadequacy of activation quantization scales in the co-training framework. To address this, the authors propose a Task-Specific Scales Quantization method suitable for multi-task co-training.

**Strengths:**

1. This work effectively incorporates quantization-aware training into co-training and significantly reduces the performance gap between multi-task co-trained models and their 4-bit quantized counterparts.
2. The authors design task-specific learnable multi-scale activation quantizer and SLLD to solve the issues of naive integration.
3. From the experimental results, it appears that the author's techniques are effective.

**Weaknesses:**

1. In table 1, quantitative results for super-resolution tasks are shown. But I am still curious about the results of deraining and denoising tasks.
2. How about the parameters change of your method?
3. Can you provide the some comparisons with some SOTA single task methods to further demonstrate the superiority of your method?

**Questions:**

See weakness.

---

> ### Author Response · Authors · 2024-11-15
> **Response to Reviewer LoRG**
>
> **We sincerely appreciate Reviewer LoRG’s valuable feedback and constructive comments.
> We will revise our organization and emphasize those contents in the main paper to better highlight the effectiveness of our method. We address each question below.**
>
> **Q1:**
> In table 1, quantitative results for super-resolution tasks are shown. But I am still curious about the results of deraining and denoising tasks.
>
> **A1:**
> Thank you for your efforts. Our TSQ-MTC also achieves the best performance on denoising and deraining tasks, where detailed results can be found in Appendix Section B (Table A and Table B). We compare the 4-bit IPT quantized by TSQ-MTC with Minmax, PAMS [1], LSQ [2], and **baseline-multi** [3]. For instance, our TSQ-MTC reaches 42.02dB in derain task on Rain100L dataset, while **baseline-multi** achieves only 41.89dB. We will highlight the discussion of experiments in the main paper to more clearly and accurately present the denoising and deraining results.
>
> **Q2:**
> How about the parameters change of your method?
>
> **A2:**
> Thank you for your concern.
> The additional parameter count of our TSQ-MTC is negligible.
> Since most low-bit quantization methods use layer-wise quantization for activations [4, 5], we only chose to introduce task-specific quantization scales (scalars) for the activation quantizer to adapt to different input data.
> In Appendix Section E, we present the parameter counts and FLOPs of the models trained by **baseline_multi** and TSQ-MTC, along with their corresponding performance.
> As shown in Table F, TSQ-MTC outperforms **baseline_multi** while maintaining a nearly identical parameter count (14.38M with a compression ratio of 87.5% vs. 14.34M with 87.6%), with only a slight increase of 0.04M. This result underscores the efficiency of our method.
> We will also add these descriptions to the revised version of our paper to highlight the advantages of TSQ-MTC in model compression.
>
> **Q3:**
> Can you provide the some comparisons with some SOTA single task methods to further demonstrate the superiority of your method?
>
> **A3:**
> Thank you for your suggestion. We compare our 4-bit TSQ-MTC to several state-of-the-art single-task super-resolution quantization methods, such as PAMS [1], CADyQ [5] and QuantSR-T [6] in Appendix Section B (Table C). Our method demonstrates a clear performance advantage over single-task quantized super-resolution models. For example, in the $\times 2$ super-resolution task on the Urban100 benchmark, TSQ-MTC achieves a PSNR of 33.67dB, notably outperforming QuantSR-T, which reaches only 32.20dB.
> We will also polish our descriptions and highlight these comparisons to the revised version.
>
> **Thank you once again for the insightful feedback.**
>
> **REFERENCES:**
>
> [1] Pams: Quantized super-resolution via parameterized max scale. ECCV'2020. \
> [2] Learned step size quantization. ICLR'2019. \
> [3] Lsq+: Improving
> low-bit quantization through learnable offsets and better initialization. CVPR Workshop'2020. \
> [4] Nonuniform-to-uniform quantization: Towards accurate quantization via generalized straight-through estimation. CVPR'2022. \
> [5] Cadyq: Content-aware dynamic quantization for image super-resolution. ECCV'2022. \
> [6] Quantsr:
> accurate low-bit quantization for efficient image super-resolution. NeurIPS'2024.

---

### Author Response · Authors · 2024-11-25
**Paper Revision**

We sincerely thank all the reviewers for their constructive feedback and kind support. \
We have revised our submission based on the provided suggestions, with the updated content highlighted in blue.

Specifically, some key modifications are as follows:
- At the beginning of the experimental section, we have outlined the organization of experiments in the main paper and appendix to facilitate locating all key details (e.g., the parameter count and FLOPs comparison). (lines 397–402)
- We have referenced supplementary experiments from the appendix in the main paper (e.g., denoising and deraining experiments). (lines 429–431)
- We have corrected the writing error pointed out by Reviewer icGh. (lines 136–138)
- We have included the experimental results of Q-ViT in both the main paper and the appendix. (Table 1, Table A, and Table B)

We have adjusted the font size of the tables in the manuscript to comply with the maximum page limit while ensuring that all content remains unchanged, except for the revised sections. Additionally, we have polished some writing mistakes in the manuscript. We hope that the revised version meets your expectations.

---

### Meta-Review · Area_Chair_gKdb · 2024-12-20

**Metareview:**

In this paper, the authors presented a new low-bit quantization method for multi-task co-training. Specifically, performance degradation was observed from the study, which identified the primary issue with existing quantization-aware training (QAT) methods. Following that, the authors propose task-specific scales quantization for multi-task co-training (TSQ-MTC) to address the mismatched quantization scales. A task-specific learnable multi-scale activation quantizer and a structure-based layer-by-layer distillation were introduced. Experimental evaluation showed the effectiveness of the proposed method. The main strengths of this paper are:
- The proposed method effectively addresses the scale discrepancies among multi-task features, incorporates quantization-aware training into multi-task co-training, and significantly reduces the performance gap. The approach also intuitively makes sense.
- Novel approaches were introduced to address the studied problem. Specifically, the above-mentioned learnable quantizer and distillation components enhanced the representational ability of shared features across different tasks.
- Comprehensive experimental evaluation was provided, showing the effectiveness of the proposed method, and providing insights for following research in this direction.
- The paper is well-written, well-motivated, and easy to follow.

The main weaknesses of this paper include:
- Some concerns about the applicability of the proposed method, e.g. to other tasks such as draining and denoising, comparison to single task methods, and high-level visual tasks.
- The performance improvement of the proposed structure-based layer-by-layer distillation is incremental.
- Missing comparison to related work, e.g. most recent QAT methods.

During the rebuttal phase, most of the major concerns were well addressed. Considering the above contributions introduced by this paper, the AC is happy to recommend an Accept, but recommends the authors incorporate all the further provided evidence and clarifications into the final version.

**Additional Comments On Reviewer Discussion:**

The authors and reviewers had a good discussion during the rebuttal phase, in which the authors provided detailed responses to the concerns and questions raised by the reviewers, and the reviewers checked them and acknowledged the further evidence provided. During the AC-reviewers discussion phase, no further concerns were raised.

Overall, this paper received a consistent positive recommendation (1 Accept and 2 borderline Accept) from the reviewers and the concerns raised in the initial review phase were addressed in the later discussions. The final decision was made mainly based on the above points.

---

### Decision · Program_Chairs · 2025-01-22

Accept (Poster)